# Research on Influencing Mechanism of Fashion Brand Image Value Creation Based on Consumer Value Co-Creation and Experiential Value Perception Theory

Lihong Chen [1] , Habiba Halepoto [2] , Chunhong Liu [1],*, Xinfeng Yan [3] and Lijun Qiu [1]

[1] Shanghai International Fashion Science and Innovation Center, Donghua University, Shanghai 200051, China; lhckxyy@dhu.edu.cn (L.C.); qiulijun20180916@163.com (L.Q.)
[2] Engineering Research Center of Digitized Textile and Fashion Technology (Ministry of Education), Donghua University, Shanghai 201620, China; 317111@mail.dhu.edu.cn
[3] International Cultural Exchange School, Donghua University, Shanghai 200051, China; yanxf@dhu.edu.cn
* Correspondence: chliu@dhu.edu.cn

**Abstract:** In view of the current lack of fashion brand competitiveness and innovation in China, this paper puts forward the concept of fashion brand image value creation and analyzes it from five dimensions: fashion brand image design, image publicity, brand aesthetics, brand charm, and brand function. This paper explores the relationship between fashion brand image value creation, customer participation behavior, experience value perception, intention, trust, and loyalty based on consumer value co-creation and experience value perception theories. On this basis, the structural equation model is used to test the research hypothesis empirically. An online survey questionnaire was subsequently developed and conducted to verify validity and reliability by statistical analysis. The results show that the value creation of fashion brand image will positively impact brand loyalty. Customer participation behavior and experience value perception play an intermediary and chain intermediary role, and customer participation willingness and fashion brand trust play a regulatory role. This study provides new ideas and references for the value creation of fashion brand image and provides quantitative scientific data for fashion enterprises to grasp the direction of brand image value creation and implement brand construction and marketing strategies.

**Keywords:** value creation; brand loyalty; customer participation behavior; experience value perception; structural equation model

## 1. Introduction

With the development of technology and the transformation of market competition, garment enterprises have realized the importance of branding. However, while the garment industry is developing towards "branding", it faces some problems, such as low competitiveness, insufficient innovation and creativity, and insufficient brand value promotion space utilization [1,2]. Therefore, it is in the interests of individual garment businesses to enhance brand competitiveness, innovation and creativity and create higher brand value [3,4]. Currently, fashion brand image value creates a direction for fashion enterprises to address these issues [5]. Fashion brand image value creation can meet the new needs of consumers and enable customers to identify the differences between brands and product categories to improve the competitiveness of brands, and finally bring more significant economic benefits to enterprises, prolong their life and increase the value of brands [6]. Therefore, to enhance brand competitiveness and brand value, the creation of fashion brand image value has been a key topic of common concern in business and academia [7–9]. At the same time, enterprises or brands will also encounter many challenges when creating brand image value [10]. For example, brands do not pay enough attention to brand aesthetics, which leads to low brand image value innovation ability and reducing customer stickiness.

In the process of brand image value creation, due to the lack of a correct direction of value creation activities, fashion brand image value creation is poor, customers' perception of brand image value creation is low, and finally reduces brand loyalty [11–13].

Previous research on fashion brand value creation focused on the use-value of brand products, but the space for value creation of product use-value was limited. In order to create a more significant value space, fashion brand value creation should be quickly transferred to more personalized service provision, customer experience, and co-creation. Therefore, this paper explores the influencing mechanism of fashion brand image value creation on brand loyalty based on consumer value co-creation and experiential value perception theory to provide quantitative scientific data for fashion enterprises to conduct the brand image value creation and marketing strategies. Keeping this idea in mind following questions were explored through this research.

1. What effect does the value creation of fashion brand image, customer engagement behavior, and perceived value of fashion brand experience have on brand loyalty?
2. What impact does fashion brand image value creation have on customer participation?
3. What impact does customer participation behavior have on the perceived value of fashion brand experience?
4. What role do the customer participation behavior and apparel brand experience value perception play in influencing brand image value creation and brand loyalty?
5. What role does the customer participation intention play in influencing brand image value creation on customer participation behavior?
6. What role does the fashion brand trust play in influencing customer engagement behavior and the perceived value of fashion brand experience on brand loyalty?

This research would be beneficial for textile and fashion marketers to understand the influencing mechanism of image value creation of Chinese fashion brands based on consumer value co-creation and experiential value perception theory. Also, it would be helpful for academia to understand its influence more deeply.

This introduction has shown that from the point of view of fashion brand image, increasing fashion brand image value creation is preferable, while knowing which factors and how to influence fashion brand image value creation is also important. In the next sections, we have reviewed the value creation of fashion brand image, experiential value perception, consumer value co-creation, and participation behavior in order to see which factors could contribute to achieving a larger portion of fashion brand image value creation. Furthermore, the research hypotheses and theoretical model have been established in Section 3, to explain and influence the mechanism of fashion brand image value creation and brand loyalty in Section 4, as empirical tests. The article concludes with a discussion of the findings in Section 5, their implications and suggestion, and future research directions in Section 6.

## 2. Literature Review

### 2.1. Value Creation of Fashion Brand Image

Brand value creation is an important research content in the field of fashion brands. High-value brands can reduce marketing costs and bring premium income to enterprises, so enterprises are increasingly concerned about brand value creation. France et al. believe that brand value creation is to satisfy customers' pursuit of higher value with new brand value [14]. Duan and Qu believe that brand value creation results from enterprises' brand investment [15]. Jayaraman and Luo proposed that brand value creation is a means for enterprises to establish a competitive advantage through brand power, brand image, and reputation [16]. Since the ultimate goal of brand value creation is to enable customers to form a unique brand experience. The content of brand value creation includes products, services, innovation, brand image, brand relationship, etc. [17]. Based on scholars' research on brand value creation, it can be found that the connotation of brand value creation has not been clearly defined. Therefore, fashion brand image value creation refers to a series of innovative and creative activities produced by enterprises that aim to create additional

brand value based on the original limited value space to meet customers' higher goal value pursuit and maximize brand image value.

Brand image is indispensable in creating brand value, so this paper believes that the connotation of value creation of fashion brand image can be defined with the help of the connotation of brand value creation. The connotation of the fashion brand value creation can be defined as the fashion brand or enterprise using all aspects of innovation to create brand image, to let the brand grow based on original finite value space, to create additional brand value, to satisfy customers with the brand image value's pursuit of a higher goal, to maximize the value of the brand image.

### 2.2. Consumer Value Co-Creation

At present, many scholars believe that in value co-creation, customers are the co-creators of value, and what they co-create with enterprises is the experience value. Prahalad and Ramaswamy [18,19] point out that there are multiple interaction points between consumers and enterprises, enabling them to achieve value co-creation and a personalized experience. Zaborek and Mazur [20] believe that value co-creation is a positive interactive process between consumers and enterprises. Consumers actively contribute their wisdom and labor and cooperate with enterprises to invent, design, and provide valuable products, services, and experiences for other consumers. Andreu, Sánchez et al. [21] believe that consumer value co-creation broadly means consumers lead value creation activities.

Similarly, Pongsakornrungsilp and Schroeder [22] point out that the dominant player in value co-creation is consumers, and value is reflected in the experience of consumers. Consumer value co-creation can also be understood as customer participation. For example, Nambisan and Baron [23] point out that customers can satisfy their needs for information, emotion, and other content by participating in product innovation to realize value co-creation. Auh et al. [24] believe that customer participation value co-creation is an active customer participation behavior. Customers contribute knowledge, experience, and resources in participation, which can provide value to themselves and enterprises at the same time. Payne et al. [25] put forward that customer participation is the critical factor in value co-creation, reflecting the contribution degree of customers in value co-creation. To sum up, the connotation of consumer value co-creation includes the concepts of "consumers as value co-creators" and "customer participation". Therefore, the text defines consumer value co-creation as consumers, as the subject of value creation, who participate in value creation activities and create value together with enterprises.

Marcos and his peers [26] believed that engagement is the source of experience and the primary way to realize value co-creation. Zhang et al. [27] show that the primary expression mode of value creation is interaction, and both the interaction between consumers and enterprises and among consumers can bring good experience value to customers. Luo and coworkers [28] proposed that customer participation value co-creation is the premise for customers to obtain co-creation experience value. Based on the viewpoints of the above scholars, it can be known that the essence of co-creation is experience value, and the premise of producing experience value is customer participation or interaction. Based on this, Mandlik and Kadirov [29] proposed a mechanism model of consumer value co-creation, including anamorphic, process, and result. Antecedent variables are composed of factors that affect customer participation value co-creation. These variables will influence the experience value by influencing customer value co-creation behavior and finally impacting customer loyalty. Moise et al. [30] pointed out that customer participation in value co-creation, perceived risk, unique needs, control desire, and organizational support as motivation variables will affect customer participation in value co-creation behavior. Finally, customers can gain unique consumption experiences or value perception in value co-creation, and enterprises can gain brand loyalty.

Based on the studies of the above scholars, the internal mechanism of consumer value co-creation conforms to the "motivation-process-result" model. Among them, customer participation in value co-creation varies according to different research fields, and the

motivation will impact customer participation in value co-creation (customer participation behavior). Experiential value or perceived value can be either a process variable or a result variable of customer participation in value co-creation, but it is always produced by value co-creation. Customer and brand loyalty are the direct result variables of consumer value co-creation.

### 2.3. Customer Participation Behavior

Existing research on consumer value co-creation primarily focuses on the level of customer participation behavior. Dai and Gu [31] define the connotation of customer participation from different research fields: Consumer participation refers to the specific behavior in which customers help create value in participating in products and services based on the traditional consumption field. Based on virtual community, customer participation behavior is a dynamic behavior of forwarding, sharing, and leaving comments. However, both emphasize the initiative of consumers or users in the process of participation. Growth [32] believes that customer participation is customers' behavior in service production and delivery, also called customer cooperative production behavior. Bove [33] states that customer participation behavior is a necessary consumer value co-creation behavior, and enterprises expect this behavior. Based on the above analysis, the connotation of customer participation behavior is defined as the behavior that consumers must take to successfully realize value creation in producing and delivering products or services.

Scholars have studied the constitutive dimensions of customer participation behavior. Ennew and Binks [34] proposed that customer participation is essentially the behavior and process of customer participation in value creation, mainly including three dimensions: information sharing, responsible behavior, and interpersonal interaction. Yi and Gong [35] believed that information sharing should be based on information search and thus divided into four dimensions of consumer participation behavior: information search, information sharing, responsible behavior, and interpersonal interaction. Bu Qingjuan et al. [36] divided customer participation behaviors in virtual communities into help-seeking, interpersonal interaction, feedback, and advocacy. Wu and Chen [37] divided customer participation behavior into three dimensions: information sharing, cooperation, and joint decision making according to the degree and process of customer participation. Participation behavior in consumer value co-creation behavior focuses on the interaction behavior between consumers and enterprises. Therefore, this paper chooses responsibility behavior and interpersonal interaction as the dimensions of customer participation behavior. In addition, the willingness of consumers to participate in value co-creation is the prerequisite for the occurrence of participation behavior, and it is influenced by the information provided by brands to consumers. Therefore, information search and information sharing are also considered the dimensions of customer participation behavior in this study.

### 2.4. Experiential Value Perception

Experiential value perception is a new concept that combines customer value perception, experiential value theory, and customer consumption behavior. The experience value perception comes from value co-creation and interaction between customers and brands [38]. Consumers' perceived experience value refers to consumers' perceived preference and evaluation of branded clothes and products [39]. The experience value is the cognition and evaluation of customers on enterprise services and advertisements based on their perception [40], which tends to be a psychological feeling formed by the interaction of experience and feeling on many vital points. This study proposes that experience value perception is consumers' comprehensive feeling and evaluation of many experience elements in value creation or other activities.

Scholars in different research fields have different classifications of perceived dimensions of experiential value. Kim and Oh [41] point out that the corresponding dimensions of experiential value perception are functionality, emotion, and social and experiential value perception in social networks. For mobile data services, experience value perception is

divided into practicality and hedonic experience value perception. Huang [42] believes that customers' experience value perception includes functional experience value perception and emotional experience value perception in the social field of the mobile short video. This paper holds that the experiential value perception under consumer value co-creation manifests as practical, functional, and social experiential value perception. The connotation of emotional experience value perception can be defined as an inner feeling generated by consumers' processing and analysis of perceived information through thinking activities such as association based on their own experience and experience. Specifically, it refers to the value of feelings such as "fun" and "relaxation" obtained in the process. Functional experience value perception refers to consumers' perception of functional utility, shown by the brand's inherent attributes and primary performance. Social experience value refers to the customer's perception of the brand image value or symbolic value during the experience, such as the brand's social status, social reputation, and other content.

## 3. Hypotheses and Model Establishment

### 3.1. Research Hypotheses

The high quality of garments impacts the buyer's choice, particularly knowing the raw material source [1]. A high-value brand image can bring higher consumer loyalty and brand premium to a brand. Chen and Zhang [43] believe that customer-centered and value-oriented brand image value creation has successfully created brand awareness, reputation, and loyalty as brand value creation can affect consumers' cognition, trust, and loyalty to brands through value co-creation. Joshi and Sharma [44] believed that brand value creation based on consumer participation would directly affect customers' repeated purchase behavior. Dai et al. [45] point out that value creation can help brands attract more customer groups and gain higher brand loyalty. Brand value creation includes many aspects, among which the value creation of the brand image is one of them, and it can bring positive effects to enterprises, including price premium, customer satisfaction, and brand loyalty [46]. In conclusion, brand image value creation has a positive impact on brand loyalty; therefore, Hypothesis H1 is proposed in this paper:

**Hypothesis H1.** *The value creation of fashion brand image has a positive effect on brand loyalty.*

Enterprises will transmit various resource information to consumers in value creation, including corporate values and corporate culture connotation [47]. This will have an impact on customer participation in product development and upgrading. Consumers began to actively participate in the process of value creation through the internet [48], during which the value creation of enterprises' products or services would promote the participation of consumers, and such customer participation behavior is reflected in the contribution of customers' knowledge, skills, and experience in the consumption field. Wu and Chen [37] pointed out that consumers play value creators, and value co-creation with the brand is inseparable from the brand's offerings. Brand elements' design and production activities and other aspects will affect the consumer's participation behavior [1].

Similarly, in the process of creating the value of fashion brand image, the realization method of creating the value of fashion brand image will increase the impression of the brand image in consumers' minds, enhance consumers' perception of the brand image, and thus promote the creation of consumers' value co-creation behavior. For example, differentiated marketing can meet the unique needs of customers. The customer's unique demand is an essential motivation for participating in value co-creation and significantly influences customers participating in value co-creation.

To sum up, value creation will influence customers' participation behavior or value co-creation behavior to contribute their knowledge and experience through value transmission. Accordingly, Hypothesis H2 is proposed in this paper:

**Hypothesis H2.** *Fashion brand image value creation has a positive impact on customer participation.*

When analyzing the relationship between customer participation behavior and brand loyalty, customer participation behavior in participating in value co-creation will enhance customers' understanding of the enterprise, thus promoting brand loyalty [49]. Based on the co-creation theory of consumer value, Hoyer and coworkers proposed that customer engagement enhances brand loyalty by improving the brand experience [50]. Etgar [51] points out that consumers can have a high level of interaction with enterprises through cooperative behavior and information sharing (cooperative behavior and information sharing are the manifestations of customer participation behavior). Consumers enjoy more autonomy and control during this period to satisfy their personalized needs and preferences, thus enhancing brand loyalty [52]. Experience-based consumer value co-creation is a high level of customer participation mode, providing innovative thinking for enterprises and positively influencing consumer loyalty [16,19,27]. Thus, customer participation in consumer value co-creation will promote brand loyalty, and in the process of participation, it is continuously strengthened and consolidated. Accordingly, Hypothesis H3 is proposed in this paper:

**Hypothesis H3.** *Customer engagement behavior has a positive impact on brand loyalty.*

Zhao [53] believes that experience value is essentially the value of customers' subjective perception and is affected by customers' participation behavior. In the research field of value co-creation, Yang [54] proposed that customers have initiative, and their active participation can enable them to obtain unique consumption experiences and promote the generation of experience value perception. Customers' participation in value co-creation can enhance their sense of identity with enterprise value, thus enhancing experience value perception, including emotional, social, and functional experience value perception. My-Quyen [55] believes that experience value perception can be obtained from the function, efficiency, and other physical attributes of customer participation in value co-creation activities or creating certain feelings. Zhang and Chen [56] believe that customers' participation in value co-creation will generate deeper emotions for brands, namely, improving the value perception of emotional experience. Vahdat [57] and Tang [58] proposed that luxury goods are not available in luxury marketing, so customer participation behavior has a more significant impact on emotion and social experience value perception.

In conclusion, under the value co-creation mode, customer participation behavior will affect the perceived value of the fashion brand experience. Therefore, Hypothesis H4 is proposed in this paper:

**Hypothesis H4.** *Customer participation behavior has a positive impact on the perceived value of fashion brand experience.*

There are many driving factors of brand loyalty, among which is customer perceived experience value. Paulose and Shakeel 2021 [59] believe consumers are loyal to brands with high perceived experience value. From the perspective of customer participation value co-creation, experience value perception is affected by customer participation behavior, affecting customer loyalty to the brand. Zhao [53] points out that when customers participate in value co-creation, their emotional experience value perception and recreational experience value perception are strengthened, and their feelings towards products are deepened. As a result, they have a higher brand commitment and loyalty and actively promote and recommend brands and products to others. Zhang and coworkers [60] pointed out that customers' perceived emotional and functional experience value would promote their trust and loyalty to the brand. Guan [61] points out that the value of the social experience will significantly meet consumers' psychological and spiritual needs and then promote consumers to have a strong cognition, association, and loyalty to the brand. It can be seen from the above that the perceived value of fashion brand experience will have a positive impact on brand loyalty. Therefore, Hypothesis H5 is proposed in this paper:

**Hypothesis H5.** *The perceived value of fashion brand experience has a positive impact on brand loyalty.*

In the process of customer participation in value co-creation, enterprises or brands make full use of the valuable resources brought by consumers through the participation of consumers, complete the process of value creation and delivery, and obtain economic returns and customer satisfaction loyalty. Nardi et al. [62] proposed that brand value creation could affect consumers' cognition and attitude towards the brand through customer participation, including customer satisfaction, trust, and loyalty. Therefore, it can be inferred that customer participation during value co-creation plays an essential role in the relationship between value creation and brand loyalty. Chen [63] points out that brands will guide brands to grasp the consumption trend through accurate customer information exchange, information sharing, and other participating behaviors conducive to improving customers' brand recognition, association, satisfaction, and loyalty in value creation activities. Focusing on introducing new materials in the garment industry or proposing logistics models for the supply chain has been practiced for a long time [64,65]. The enterprise or the brand value creation pattern changed under value-creating behavior or participation behavior [27]. They maintained that primary performance for our customers to participate in the brand product research and development, production, logistics, and marketing leads to value creation concerns from use-value to the experience value transfer, thereby enabling customers to form a higher quality of brand loyalty. In conclusion, brand value creation will promote brand loyalty under the influence of value co-creation behavior or customer participation behavior. Therefore, Hypothesis H6 is proposed in this paper:

**Hypothesis H6.** *Customer participation plays a mediating role in the relationship between the value creation of fashion brand image and brand loyalty.*

With the coming of the era of the experience economy, value creation is rapidly shifting to customer experience, and customer participation in the behavior of value co-creation is the premise of customer experience value perception. Teng and Tsai put forward from tourism management that the value creation of tourism brands will affect tourists' perception of tourism experience value through their participation behavior [66]. For example, when tourists join the brand's interactive social activities, their participation behaviors such as information exchange and sharing will impact the value perception of social experience. Brand value creation under customer participation behavior would impact experiential value perception; that is, customer participation behavior has a specific mediating effect on the influence relationship between brand value creation and experiential value perception [16,19,67]. From experiential marketing, Yoo et al. [68] point out that brand value creation must rely on certain material carriers and bring different experiential value perceptions to customers through their participation behavior. Therefore, in creating the image value of a fashion brand, the carriers of its experience value are image design, image publicity, brand aesthetics, brand charm, and brand function, which will impact the perception of experience value through the customer's participation behavior.

Experiential value perception significantly affects the relationship between participation behavior, "information exchange and interpersonal interaction", and tourist loyalty. Since customer participation in value co-creation can impact customer loyalty through experience value perception. Fang et al. [69] believe that user participation in the context of value co-creation can benefit both customers and brands. Customers' participation behaviors (comments, inquiries, and interpersonal interactions with other consumers) in the online shopping environment can improve customers' perception of experience value, thus improving their repeated purchases, recommendation, and other behaviors [6,70]. In conclusion, customer participation behavior positively impacts brand loyalty, and the perceived value of fashion brand experience plays a significant role in this influencing relationship. In other words, customer participation behavior can positively affect brand loyalty through the perceived value of the fashion brand experience.

Therefore, the image value creation of fashion brands can make customers perceive the experience value of fashion brands through customer participation behavior and positively impact brand loyalty. Therefore, Hypothesis H7 is proposed in this paper:

**Hypothesis H7.** *Customer participation behavior and apparel brand experience value perception play an intermediary chain role in brand image value creation affecting brand loyalty.*

The willingness of customers to participate in this paper refers to the willingness of consumers to participate in value co-creation, which originates from consumers' cognition of value creation activities to a certain extent and is the basis for participating in value co-creation [71]. Venkatesh and Davis [72] believe that consumers' willingness to participate directly influences actual participation behavior. In the context of customer participation, it should be noted that the "willingness" is customers' attitude toward their participation in product creation and shows in her research that willingness can regulate consumers' behavior of participating in value creation. When consumers are highly willing to participate, they want to participate in value-creation activities. The customers usually gradually generate the willingness for value co-creation in the interactive atmosphere created by enterprises and then produce value co-creation behavior. Based on the above analysis, the willingness of customers to participate in value co-creation can promote customer participation behavior. Therefore, Hypothesis H8 is proposed in this paper:

**Hypothesis H8.** *Customer participation intention plays a moderating role in the influence of brand image value creation on customer participation behavior.*

Gavilan [73] points out that customers' trust in hotels is mainly based on other consumers' online participation behaviors. For example, online reviews of hotels will affect customers' trust in the hotel brand to some extent. Brand trust is a process of regulating the influence of customer participatory value co-creation on brand loyalty through the level of customer perceived risk and perceived value. When customers' perceived risk or value is higher, their desire to participate in the value co-creation will be stronger, leading to more active participation behavior of customers to enhance brand loyalty. After the occurrence of participatory behavior (value co-creation behavior), customers will have a psychological evaluation of the brand, namely brand trust, and thus affect consumer loyalty [35,38]. Azize [74] believes that users' shared and interactive participation in the brand community will increase brand loyalty, and users' trust in the community plays a moderating role. Thus, consumers' participation in value co-creation can deepen their understanding of enterprises and help them establish emotional links with enterprises. Moreover, under the influence of brand trust, consumers' repeat purchase behavior and recommendation intention will be further enhanced.

Based on the above analysis, fashion brand trust plays a facilitating role in the relationship between customer participation behavior and brand loyalty. Therefore, Hypothesis H9 is proposed in this paper:

**Hypothesis H9.** *Fashion brand trust plays a moderating role in influencing customer engagement behavior on brand loyalty.*

Consumer experience value perception results from a comprehensive feeling, which is essential for understanding the customer-brand relationship. For example, in the virtual brand community, only when consumers first form trust in the brand community will they take the initiative to deeply understand the brand products and services in the community to obtain the perception of experience value and finally form brand loyalty. Elena and Jose [12] believe that value perception is influenced by trust, and customers with high brand trust will get more perceived benefits and value than other consumers, and it is easier to form brand loyalty. The experience value perception generated by customer value co-creation behavior will positively impact brand loyalty through brand trust. In addition, the

level of consumer trust is positively correlated with the value perception, and the interaction between the two will further determine brand loyalty. The consumers' level of trust in brands will affect their access to brand resource information, thus affecting their perception of experience value and ultimately influencing consumers' brand attachment [45].

In conclusion, the relationship between perceived experience value and brand loyalty is disrupted by brand trust, and when consumers form brand trust, they are more likely to generate brand loyalty. Therefore, Hypothesis H10 is proposed in this paper:

**Hypothesis H10.** *Fashion brand trust plays a moderating role in influencing the perceived value of fashion brand experience on brand loyalty.*

*3.2. Theoretical Model Establishment*

Taken together, this theoretical model is established based on consumer value co-creation theory and experiential value perception theory. This is achieved by combining the above research hypotheses, this paper takes the customer participation behavior and the perceived value of apparel brand experience as the intermediary variable and the customer participation willingness and brand trust as the moderating variable and constructs a theoretical model of the impact of apparel brand image value creation on brand loyalty, as shown in Figure 1.

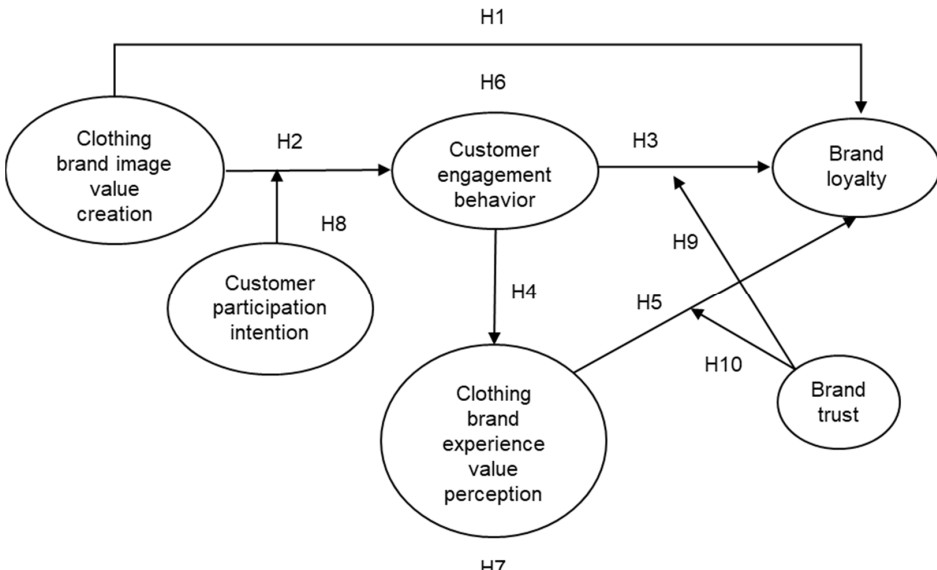

**Figure 1.** The theoretical model for this study.

## 4. Empirical Test and Data Analysis

*4.1. Item Development*

The measurement scale in this paper is based on the relevant mature scale to be screened and modified. Among them, customer participation behavior mainly borrows from the scale of Yi and Gong [35]. The perceived value of fashion brand experience is mainly based on the scale of Shin [75] and Hennigs [76]. Brand loyalty is mainly based on the measurement scale of Guan [61] and Apenes Solem [11]. Customer participation intention is mainly based on the scale of [53]. Brand trust is mainly based on the scale of Elena and Jose [12].

There is no mature fashion brand image value creation scale, so it is necessary to develop a measurement scale. This paper constructs an index system of fashion brand image value creation based on the grounded theory, which includes five dimensions, including image design, image publicity, brand aesthetics, brand charm, and brand function, and

29 indexes. Based on much literature and previous studies, a suitable measurement scale was formed through modification and adjustment, as shown in Table 1.

**Table 1.** Measurement scale of fashion brand image value creation.

| Dimension | Number | Secondary Index | Item | Reference |
|---|---|---|---|---|
| Image design | ID1 | Product image design | Product image design novel. | Cai [4] Marine [7] Cai and Cheng [77] |
| | ID2 | Store image design | Store image design is unique. | |
| | ID3 | Brand identity image design | Brand identity image design has identification. | |
| | ID4 | Packaging design | The packaging design is simple and generous. | |
| | ID5 | Price image design | Reasonable price image design. | |
| Image publicity | IP1 | Designer image publicity | The brand designer image is consistent with the designer image in my mind. | Fu and Cao [78] |
| | IP2 | Advertising image promotion | Brand ads usually get my attention. | |
| | IP3 | Service image publicity | Branded service usually satisfies me. | |
| | IP4 | Corporate image publicity | The brand that belongs to the enterprise has a good reputation. | |
| | IP5 | Customer image promotion | Customers of brands usually have a good personal image. | |
| | IP6 | Promotion image publicity | The promotional activities of the brand can continually deepen my impression of the brand. | |
| Brand Aesthetics | BA1 | Brand style | The brand style is usually popular with the public. | Godey [79] Cai [4] |
| | BA2 | Brand symbol | Brand symbols are usually aesthetic. | |
| | BA3 | Brand color | The use of brand color is usually in line with the public aesthetic concept. | |
| | BA4 | Brand design | Brand patterns are usually representative | |
| | BA5 | Brand poster | Brand posters are usually unique and original. | |
| | BA6 | Brand store atmosphere | The store atmosphere is usually pleasant and comfortable. | |
| Brand Charm | BC1 | Brand concept | The brand concept is in line with the public concept | Koenigsberg [80] |
| | BC2 | Brand spirit | The brand spirit is positive | |
| | BC3 | Brand culture | Profound brand culture | |
| | BC4 | Brand reputation | Good brand reputation | |
| | BC5 | Brand personality | Strong brand personality | |
| | BC6 | Brand added value | The brand has added brand value. | |
| | BC7 | Brand value orientation | The public can recognize brand value orientation | |
| Brand function | BF1 | Recognition function | This brand makes it easy for me to distinguish myself from other brands. | Ma [81] Jiao [8] Gu and Xu [82] |
| | BF2 | Quality commitment and assurance function | The brand's products are of reliable quality. | |
| | BF3 | Communication and shopping guide function | The brand can convey the brand or product message to me. | |
| | BF4 | Competition function | The brand has a highly competitive advantage | |
| | BF5 | Value chain function | The brand will usually cooperate with well-known brands | |

## 4.2. Data Collection

The questionnaire survey was used to collect data, and online questionnaires were distributed through the "Questionnaire Star" software. As a principal target to investigate the perceptions of fashion brand image value creation by diverse types of consumers, all consumers who had ever purchased branded clothing via physical or internet shops were the subject of our research. For the purpose of matching the subjects' characteristics, we asked consumers to choose what brands they were concerned about. A total of 642 questionnaires were issued, of which 567 were valid, with an effective rate of 88.3%. In order to ensure the reliability and validity of the questionnaire, this study first conducted a pre-survey within a small group, and this questionnaire was partially modified based on the pre-survey results. The descriptive statistical results of the final survey samples collected are shown in Table 2. It can be seen from the table that all ages, occupations, regions, and monthly incomes were involved in the survey, indicating that the selection of survey groups was reasonable.

**Table 2.** The basic information of survey samples.

| Statistical Variables | | Sample Size | Proportion (%) |
|---|---|---|---|
| Gender | Male | 270 | 47.62% |
| | Female | 297 | 52.38% |
| Age | $\leq$18 | 19 | 3.41% |
| | 19–25 | 185 | 32.68% |
| | 26–30 | 157 | 27.63% |
| | 31–40 | 110 | 19.34% |
| | 41–50 | 70 | 12.35% |
| | $\geq$51 | 26 | 4.59% |
| Occupation | Party and government officials | 41 | 7.23% |
| | Professional and technical personnel, doctors | 53 | 9.35% |
| | Teachers | 63 | 11.11% |
| | Company management cadre | 30 | 5.29% |
| | Company worker | 115 | 20.28% |
| | Students | 184 | 32.45% |
| | Business service worker | 42 | 7.41% |
| | Self-employed | 36 | 6.35% |
| | Other | 3 | 0.53% |
| Education background | High school and below | 55 | 9.65% |
| | Junior college | 84 | 14.88% |
| | Undergraduate | 245 | 43.26% |
| | Master's degree or above | 183 | 32.21% |
| Monthly income | $\leq$1500 RMB | 167 | 29.45% |
| | 1501–3000 RMB | 71 | 12.52% |
| | 3001–5000 RMB | 114 | 20.11% |
| | 5001–8000 RMB | 137 | 24.16% |
| | 8001–12,000 RMB | 54 | 9.52% |
| | $\geq$12,001 RMB | 24 | 4.24% |
| Place of abode | First-tier cities | 173 | 30.51% |
| | Second-tier cities | 222 | 39.15% |
| | Third line cities | 104 | 18.34% |
| | Fourth tier cities or other areas | 68 | 12.00% |

### 4.3. Reliability and Validity Test

Data testing was carried out on the measurement scale of value creation of fashion brand image, mainly including reliability and validity test, factor test, and goodness of fit test of the model. In the exploratory factor molecule, the cumulative variance contribution rate of the scale was 73.061% > 60%, and the overall KMO and Cronbach's α values were 0.946 and 0.965, respectively, both greater than 0.7. Meanwhile, the KMO and Cronbach's α values of each dimension also reached the acceptance standard. In the confirmative factor analysis, the standardized factor loading (EFA) was between 0.648 and 0.912, meeting the criteria of greater than 0.5, and the combinatorial reliability (CR) and square extraction variance (AVE) also met the criteria of 0.7 and 0.5. In the goodness of fit test of the model, the chi-square to the degree of freedom ratio (X2/DF) is 2.898, which is less than 3, AGFI and GFI are more significant than 0.8, and NFI, CFI, TLI, and IFI are all greater than 0.9. The measurement scale has good reliability and validity based on the above analysis.

SPSS25.0 was used for reliability and validity tests and confirmative factor analysis in this study. The specific results are shown in Table 3. Cronbach's α coefficient and KMO value of each measurement variable meet the accepted standard of the lowest 0.5, and Cronbach's α coefficient and KMO value of the whole model are more significant than 0.7, which can be considered good reliability and validity of the model. In addition, the combined reliability and mean-variance of all the measured variables were within the acceptable range, and the confirmatory factor analysis results were good.

**Table 3.** The test of model reliability and validity and confirmatory factor analysis.

| Dimension | | Measurement Item | CITC | α After Deleting | α | KMO | Bartlett's Test of Sphericity | | | EFA | CR | AVE |
|---|---|---|---|---|---|---|---|---|---|---|---|---|
| | | | | | | | Chi-Square | Degrees of Freedom | Significant | | | |
| Fashion brand image value creation | | ID | 0.730 | 0.864 | 0.888 | 0.888 | 1274.11 | 10 | 0.000 | 0.775 | 0.899 | 0.642 |
| | | IP | 0.727 | 0.863 | | | | | | 0.811 | | |
| | | BA | 0.716 | 0.866 | | | | | | 0.859 | | |
| | | BC | 0.768 | 0.854 | | | | | | 0.832 | | |
| | | BF | 0.700 | 0.870 | | | | | | 0.723 | | |
| Customer engagement behavior | | CPB1 | 0.745 | 0.913 | 0.922 | 0.851 | 2291.85 | 15 | 0.000 | 0.794 | 0.923 | 0.666 |
| | | CPB2 | 0.813 | 0.903 | | | | | | 0.852 | | |
| | | CPB3 | 0.815 | 0.903 | | | | | | 0.865 | | |
| | | CPB4 | 0.771 | 0.909 | | | | | | 0.818 | | |
| | | CPB5 | 0.760 | 0.910 | | | | | | 0.777 | | |
| | | CPB6 | 0.757 | 0.911 | | | | | | 0.785 | | |
| Fashion brand experience value perception | Emotional experience value perception | AEVP1 | 0.694 | 0.871 | 0.875 | 0.694 | 831.85 | 3 | 0.000 | 0.767 | 0.883 | 0.716 |
| | | AEVP2 | 0.840 | 0.750 | | | | | | 0.916 | | |
| | | AEVP3 | 0.752 | 0.832 | | | | | | 0.849 | | |
| | Functional experience sense of value | FEVP1 | 0.802 | 0.846 | 0.892 | 0.805 | 1193.63 | 6 | 0.000 | 0.851 | 0.904 | 0.701 |
| | | FEVP2 | 0.748 | 0.866 | | | | | | 0.801 | | |
| | | FEVP3 | 0.802 | 0.844 | | | | | | 0.901 | | |
| | | FEVP4 | 0.697 | 0.883 | | | | | | 0.792 | | |
| | Value perception of social experience | SEVP1 | 0.686 | 0.863 | 0.873 | 0.695 | 820.47 | 3 | 0.000 | 0.758 | 0.881 | 0.713 |
| | | SEVP2 | 0.834 | 0.749 | | | | | | 0.913 | | |
| | | SEVP3 | 0.755 | 0.823 | | | | | | 0.854 | | |

**Table 3.** *Cont.*

| Dimension | Measurement Item | CITC | α After Deleting | α | KMO | Bartlett's Test of Sphericity | | | EFA | CR | AVE |
| | | | | | | Chi-Square | Degrees of Freedom | Significant | | | |
|---|---|---|---|---|---|---|---|---|---|---|---|
| Brand loyalty | BL1 | 0.695 | 0.861 | 0.882 | 0.814 | 1394.81 | 10 | 0.000 | 0.756 | 0.884 | 0.604 |
| | BL2 | 0.697 | 0.861 | | | | | | 0.753 | | |
| | BL3 | 0.765 | 0.844 | | | | | | 0.808 | | |
| | BL4 | 0.774 | 0.843 | | | | | | 0.828 | | |
| | BL5 | 0.652 | 0.871 | | | | | | 0.736 | | |
| Customer participation intention | CPI1 | 0.780 | 0.903 | 0.910 | 0.747 | 1081.33 | 3 | 0.000 | 0.829 | 0.911 | 0.774 |
| | CPI2 | 0.838 | 0.852 | | | | | | 0.899 | | |
| | CPI3 | 0.837 | 0.853 | | | | | | 0.909 | | |
| Brand trust | BT1 | 0.799 | 0.850 | 0.894 | 0.813 | 1210.69 | 6 | 0.000 | 0.731 | 0.850 | 0.588 |
| | BT2 | 0.730 | 0.875 | | | | | | 0.712 | | |
| | BT3 | 0.829 | 0.837 | | | | | | 0.893 | | |
| | BT4 | 0.703 | 0.886 | | | | | | 0.716 | | |

Overall KMO = 0.927, Bartlett test chi-square value = 10,153.84, Sig value = 0.000, Cumulative variance contribution rate = 74.78%, overall Cronbach's α coefficient = 0.954.

### 4.4. Goodness of Fit Test of Model

AMOS22.0 was used for the goodness of fit test of the model in this study, and the specific results are shown in Table 4. Since when X2/DF is less than 3, RMR is less than 0.05, RMSEA is less than 0.08, AGFI and GFI are greater than 0.8, and other indicators are greater than 0.9, it indicates that the model has a good degree of the fitting. It can be seen from the table that all indicators of each measurement variable meet the requirements, so it can be considered that the relevant results of each measurement model are relatively ideal.

**Table 4.** The test of model goodness of fit index.

| Fitting Index | Fashion Brand Image Value Creation | Customer Engagement Behavior | Fashion Brand Experience Value Perception | Brand Loyalty | Customer Participation Intention | Brand Trust |
|---|---|---|---|---|---|---|
| CMIN | 71.613 | 17.928 | 76.321 | 55.384 | 5.274 | 31.216 |
| X2/DF | 2.170 | 2.988 | 2.544 | 2.408 | 2.637 | 1.951 |
| RMR | 0.020 | 0.029 | 0.033 | 0.012 | 0.039 | 0.026 |
| RMSEA | 0.049 | 0.063 | 0.056 | 0.053 | 0.033 | 0.044 |
| GFI | 0.939 | 0.988 | 0.972 | 0.945 | 0.918 | 0.944 |
| AGFI | 0.911 | 0.958 | 0.948 | 0.927 | 0.859 | 0.908 |
| NFI | 0.953 | 0.992 | 0.977 | 0.946 | 0.942 | 0.913 |
| IFI | 0.956 | 0.995 | 0.966 | 0.948 | 0.945 | 0.937 |
| CFI | 0.956 | 0.995 | 0.966 | 0.948 | 0.945 | 0.937 |

### 4.5. Hypothesis Testing

The test results of the theoretical model are shown in Table 5. H1 path coefficient is 0.602, and the T value is 6.324, which meets the significant standard, indicating that the value creation of fashion brand image has a significant positive impact on brand loyalty. Therefore, Hypothesis H1 is established. H2 path coefficient is 0.863, T value is 16.182, reaching the significant standard, indicating that the value creation of fashion brand image has a significant positive impact on customer participation behavior, so the assumption of

H2 is valid. The diameter coefficient of H3 is 0.500, and the T value is 4.475, which reaches the significant standard, indicating that customer participation behavior has a significant positive impact on brand loyalty. Hypothesis H3 is established. The path coefficient of H4 was 0.876, and the T value was 18.436, which reached the significant standard, indicating that customer participation behavior had a significant positive impact on the perceived value of fashion brand experience. Hypothesis H4 was established. The path coefficient of H5 was 0.481, and the T value was 4.287, which reached the significant standard. Based on the above analysis, the perceived value of fashion brand experience significantly impacts brand loyalty, so hypothesis H5 is established.

**Table 5.** The result of a hypothesis test.

| Hypothesis | Path Coefficient | S.E. | T Value | *p*-Value | Conclusion |
|:---:|:---:|:---:|:---:|:---:|:---:|
| H1 | 0.602 | 0.107 | 6.324 | *** | Support |
| H2 | 0.863 | 0.067 | 16.182 | *** | Support |
| H3 | 0.500 | 0.116 | 4.475 | *** | Support |
| H4 | 0.876 | 0.039 | 18.436 | *** | Support |
| H5 | 0.481 | 0.123 | 4.287 | *** | Support |

Note: *** indicates that *p* is significant below 0.001.

*4.6. Mediation Effect Test*

In this study, the Bootstrap method was used to test the mediating effect of Hypothesis H6 customer participation behavior, the chain mediating effect of Hypothesis H7 customer participation behavior, and the perceived value of fashion brand experience. When testing Hypothesis H6, a simple mediation model (Model4) is used to analyze the mediation effect of customer participation behavior under controlling the basic information of the survey objects. The test results are shown in Table 6. The upper and lower limits of the bootstrap95% confidence interval do not contain 0, the direct effect value is 0.451, and the indirect effect value is 0.322, accounting for 58.34% and 41.66% of the total effect, respectively. It shows that customer participation has a significant (partial) mediating effect on the relationship between the value creation of fashion brand image and brand loyalty, and hypothesis H6 has been verified. When testing Hypothesis H7, Model6 analyzes the chain mediating effect between customer participation behavior and fashion brand experience value perception under the same control of basic information. The specific results are shown in Table 7. The upper and lower limit of the BootCI confidence interval does not contain 0, the direct effect value is 0.463, and the indirect effect value is 0.180 and 0.158, respectively. This indicates that customer participation behavior and fashion brand experience value perception have an intermediary chain role (partial intermediary) in the impact of fashion brand image value creation on brand loyalty. Hypothesis H7 is verified.

**Table 6.** Customer Participation Behavior Mediation Effect Analysis Results.

| Path | Effect Value | Boot Standard Error | BootCI Lower Limit of the Confidence Interval | BootCI Upper Limit of the Confidence Interval | Effect Proportion |
|:---:|:---:|:---:|:---:|:---:|:---:|
| Total effect | 0.773 | 0.0455 | 0.7506 | 0.8993 | - |
| Direct effect | 0.451 | 0.0515 | 0.5610 | 0.7633 | 58.34% |
| Indirect effect (fashion brand image value creation → customer participation behavior → brand loyalty) | 0.322 | 0.0393 | 0.0593 | 0.2163 | 41.66% |

**Table 7.** Chain Effect Analysis Results.

| Path | Effect Value | Boot Standard Error | BootCI Lower Limit of the Confidence Interval | BootCI Upper Limit of the Confidence Interval |
|---|---|---|---|---|
| Total effect | 0.801 | 0.0347 | 0.7323 | 0.8684 |
| Direct effect | 0.463 | 0.0455 | 0.3523 | 0.6130 |
| Indirect effect (fashion brand image value creation → customer participation behavior → brand loyalty) | 0.180 | 0.0434 | 0.0928 | 0.2638 |
| Indirect effect (fashion brand image value creation → customer participation behavior → fashion brand experience value perception → brand loyalty) | 0.158 | 0.0391 | 0.0195 | 0.1715 |

*4.7. Moderation Effect Test*

This study used a hierarchical regression model to examine the moderating effect of customer participation intention and brand trust. Under the condition of controlling the basic information of the survey objects, the interaction items of "fashion brand image value creation × customer participation willingness", "customer participation behavior × brand trust" and "fashion brand experience value perception × brand trust" were respectively introduced for regression analysis. The specific results are shown in Table 8. After introducing the interaction term, the value of the adjusted R square gradually increases, indicating that the model is better fitted. In addition, the normalization coefficient was 1.320, 1.321, and 1.472, and the T value was 7.034, 5.903, and 5.923, respectively, and the *p* was significant at the level of 0.01. This indicates that interaction items have a significant impact on customer participation behavior or brand loyalty. Customer participation willingness has a significant moderating effect on the impact of fashion brand image value creation on customer participation behavior, and brand trust has a significant moderating effect on customer participation behavior on brand loyalty. At the same time, the perceived value of fashion brand experience also has a significant regulating effect on the influence of brand loyalty. Therefore, Hypotheses H8, H9, and H10 have been verified.

**Table 8.** The results of moderating effect.

| Variable | Customer Participation Behavior | | Variable | Brand Loyalty | | Variable | Brand Loyalty | |
|---|---|---|---|---|---|---|---|---|
| | Standardization Coefficient | T Value | | Standardization Coefficient | T Value | | Standardization Coefficient | T Value |
| Gender | 0.060 | 2.098 | Gender | −0.065 | −1.797 | Gender | −0.026 | −0.786 |
| Age | −0.013 | −0.430 | Age | 0.050 | 1.365 | Age | 0.104 | 3.059 |
| Occupation | 0.003 | 0.103 | Occupation | 0.018 | 0.510 | Occupation | 0.007 | 0.219 |
| Education | −0.014 | −0.484 | Education | 0.000 | 0.002 | Education | −0.008 | −0.251 |
| Income | −0.046 | −1.560 | Income | 0.050 | 1.351 | Income | 0.026 | 0.757 |
| Residence | −0.055 | −1.908 | Residence | 0.031 | 0.868 | Residence | 0.016 | −0.472 |
| Fashion brand image value creation | 1.201 *** | 16.368 | Customer engagement behavior | 0.561 ** | 2.939 | Fashion brand experience value perception | 0.745 *** | 4.234 |
| Customer participation intention | 1.038 *** | 6.442 | Brand trust | 0.983 *** | 5.721 | Brand trust | 0.140 ** | 3.142 |
| Fashion brand image value creation × customer participation willingness | 1.320 *** | 7.034 | Customer engagement behavior × brand trust | 1.321 *** | 5.903 | Fashion brand experience value perception × brand trust | 1.472 *** | 5.923 |
| Adjust R squared change | −0.002→0.557→0.597 | | Adjust R squared change | −0.006→0.369→0.378 | | Adjust R squared change | −0.006→0.466→0.477 | |
| F value change | 0.811→78.665→82.524 | | F value change | 0.517→37.398→33.189 | | F value change | 0.517→55.206→48.94 | |

Note: *** means *p* is significant at the level of 0.001, ** means *p* is significant at the level of 0.01.

## 5. Discussions

The empirical results show that fashion enterprises can further form brand loyalty by promoting consumers' value co-creation and enhancing experience value perception when carrying out value creation activities. The SEM model was developed to establish the relationships among variables as follows in Figure 2.

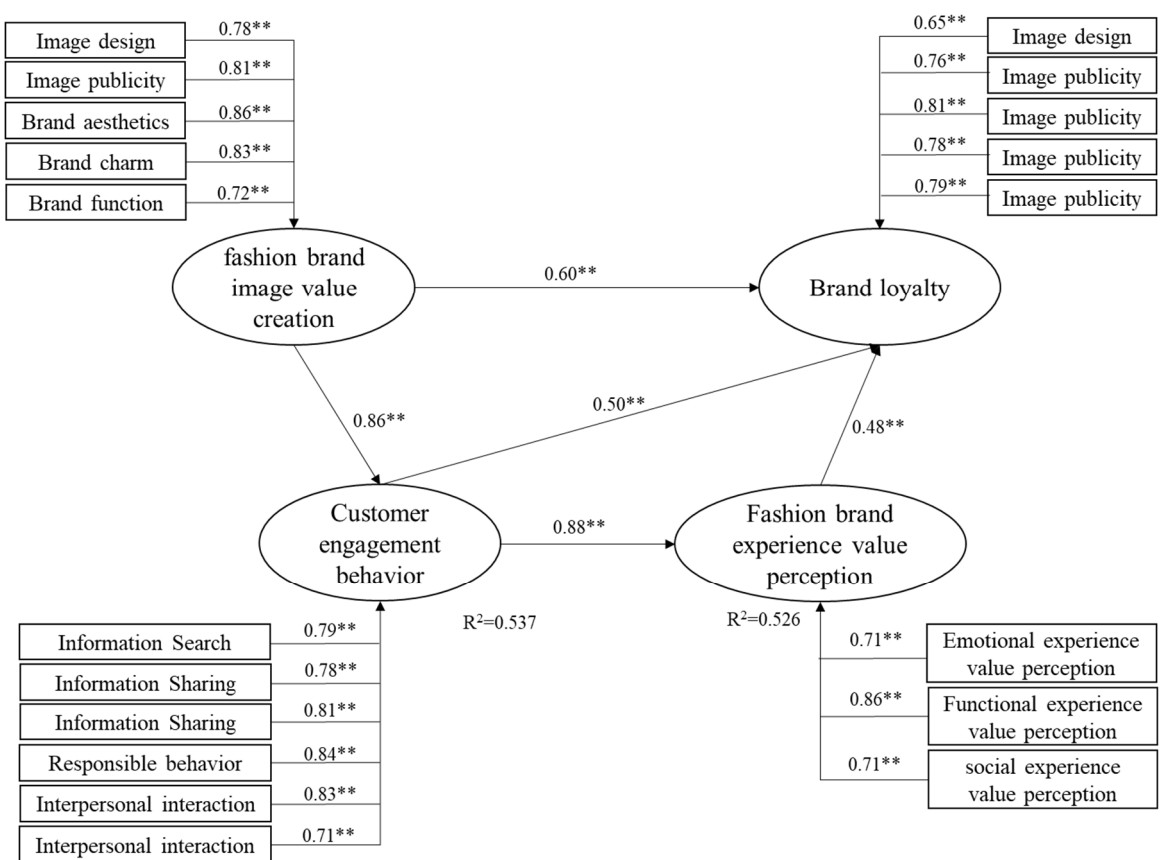

**Figure 2.** The graphical representation of the SEM model. ** means *p* is significant at the level of 0.01.

Assuming the establishment of H1 indicates that creating fashion brand image value will directly affect brand loyalty when fashion enterprises carry out value creation activities, consumers' perception of various elements of fashion brand image can be improved, thus enhancing brand loyalty. Hypotheses H2, H3, and H6 show that customer participation plays an intermediary role in the relationship between the value creation of fashion brand image and brand loyalty. When consumers' perception of various elements of the value creation of fashion brand image is high, it will stimulate consumers' interest in participating in the co-creation and then produce customer participation behavior, impacting brand loyalty. Furthermore, customer participation behavior belongs to creating the necessary link of consumer value. It can make consumers and brand both sides establish good relations of cooperation, to a certain extent, also affect the customer to the enterprise's trust, so that customers have a positive attitude orientation and enhance customer repeat purchase desire, to enhance the brand loyalty further.

It is assumed that H2, H4, H5, and H7 are established, indicating that the creation of fashion brand image value will first affect consumers' customer participation behavior and then influence brand loyalty through the role of perceived experience value, that is, customer participation behavior and perceived experience value of fashion brand have an intermediary chain role. The fashion brand experience value perception needs to be based on consumer value co-creation in fashion brand image value creation. According to the theory of consumer value co-creation, consumers' positive perception of fashion

brands will promote them to participate in the value co-creation, and the inevitable result of participating in the value co-creation is to produce a unique consumption experience, during which consumers will generate value perception of experience elements, namely, experience value perception. In addition, this perception usually leads to a favorable orientation; consumers will have a more profound emotional attachment to the brand, thus have a higher brand commitment and loyalty to the brand, and will actively promote and recommend the brand and products to the brand.

The establishment of Hypothesis H8 indicates that creating fashion brand image value on customer participation behavior is regulated by customer participation willingness. Consumers will evaluate and judge the value creation of fashion brand image based on their perception. When consumers are interested in the value creation of fashion brand image, they will hope to better experience fashion brand products or services by participating in value creation. Therefore, the degree of consumers' interest in the value creation of fashion brand image and the degree of consumers' desire to participate in the value creation is customers' willingness to participate in this influence relationship. Furthermore, the more interested they are, the stronger their hope is, their willingness to participate, and easier to generate customer participation behavior.

It is assumed that the establishment of H9 and H10 indicates that the process of customer participation behavior on brand loyalty and the process of apparel brand experience value perception on brand loyalty are both regulated by apparel brand trust. From the perspective of consumer value co-creation, consumers' trust in fashion brands will affect their behavior in value co-creation and further affect their purchasing decisions. When consumers have high trust in fashion brands, they are more active in value co-creation; Through such frequent brand interaction, consumers will have a sense of dependence on fashion brands and thus form brand loyalty. According to the theory of experiential value perception, brand trust strongly correlates with consumers' experiential value perception. When consumers trust a brand, it indicates that they have a high psychological evaluation of the brand and will actively acquire more information related to the brand or product, resulting in a positive perception of the brand, and thus promoting the generation of brand loyalty.

## 6. Conclusions

### 6.1. Theoretical Implications

This research provides a comprehensive view of the factors that influence brand loyalty: customer participation behavior, perceived experience value, brand trust, and fashion brand image value creation attributes. Customer participation causes good relations between consumers and the brand and develops brand trust through positive attitude orientation by enhancing customer repeat purchase desire and brand loyalty. Thus, brand value creation activities improve fashion brand image and enhance brand loyalty. The customer participation behavior is developed before brand loyalty through the role of perceived experience value. The value creation of a fashion brand image depends on consumers' perceptions. The consumers' interest or desire (i.e., participation willingness) of the value creation of fashion brand image is related to their hope (i.e., trust) with brands. Fashion brand image value will directly affect brand loyalty. Customer value co-creation and participation behavior on brand loyalty and purchasing decisions are regulated by apparel brand trust. Brand trust strongly correlates with consumers' experiential value perception. More brand interaction results in more brand or product information, causing higher brand psychological evaluation, which ultimately causes higher brand positive perception, higher brand loyalty, and trust in brand trust.

### 6.2. Practical Implications

In terms of promoting consumer value co-creation, it can be achieved by enhancing the interaction between customers and brands based on empirical results. First, fashion enterprises need to open interactive social platforms on websites such as Sina Weibo,

Xiaohongshu, Facebook, and WeChat to establish a social network connection between brands and consumers. Secondly, fashion enterprises should set appropriate usernames and other essential information, including corporate brand logos and related pictures in social media accounts, which is conducive to gaining the attention and trust of other users. Third, fashion enterprises need to establish an effective information feedback mechanism to obtain the consumption needs of online consumers, which requires enterprises to enhance their proactive awareness and actively release topics and activities to enhance interaction and realize value co-creation. In addition, fashion enterprises can also set up an influencers business department, focus on content marketing, and in the form of a network broadcast and soft text push to trigger the discussion among users and brands.

The primary way to increase offline interaction among consumers and brands is to use the star effect. Fashion enterprises can invite stars to participate in offline activities to attract consumers to experience the scope of brand activity atmosphere and take the opportunity to promote consumers to participate in value co-creation. Secondly, fashion enterprises can enhance the interaction with consumers by opening offline experience stores to experience the category of fashion brand concept. For example, fashion enterprises can transform fashion shows into offline experience stores with different themes in different regions. Consumers can experience the design concept and development concept of brand clothes and jointly discuss their views on the concepts to achieve value co-creation.

In terms of enhancing the perception of experience value, it can be achieved by enhancing the brand information symmetry perceived by consumers. Fashion enterprises can choose to join the virtual brand community to solve asymmetric brand information perceived by consumers. Releasing many high-quality contents related to the brand in the virtual community and setting keywords and title classification information can make consumers better understand the brand and improve it. Secondly, to enhance information symmetry, fashion enterprises should standardize the transmission mechanism of brand information and actively send "screening" information to consumers to realize the transformation of information to consumers' demands. For example, when consumers have low information perception of brand connotation, spirit, concept, and culture, fashion enterprises should promote the spiritual content of fashion brands and reduce the output of material content such as products, posters, and stores to reduce information asymmetry. For example, training offline sales personnel to explain the brand story and history can effectively improve customers' understanding of the brand connotation.

### 6.3. Limitations and Future Research

The present study has a few limitations. First, this study is limited to Chinese clothing consumers in a particular geographical region. Hence, further testing of other objects across different geographical regions is required before these results can be generalized. Second, the participants filling out questionnaires based solely on the understanding of mass consumers is not comprehensive enough and more survey samples of occupations for apparel enterprise practitioners can be added to further ensure the accuracy of the research. Third, Fashion brand image value creation is currently a relatively new concept in the field of public cognition. Through the form of questionnaires, especially in the form of online distribution, it is easy to cause some respondents to have a partially thorough understanding of the research topic and cause the conclusions to be inconsistent with the real ideas. It can also be considered to use scientific and technological means to improve the accuracy of the recovered data, or to try to use mathematical analysis methods and experimental methods to break through the limitations of research methods. Therefore, we invite future research in this area. Finally, different target consumer perceptions of fashion brand image value creation may lead to inconsistent results, thus it is recommended that other consumers from different countries be selected as subjects for investigation in the future. Further research emphasizing different subjects should be undertaken in other countries for different market-oriented targets.

**Author Contributions:** Conceptualization, L.C. and H.H.; Data curation, L.Q.; Formal analysis, L.C. and L.Q.; Funding acquisition, X.Y.; Investigation, L.C. and L.Q.; Methodology, L.C. and C.L.; Project administration, C.L.; Resources, C.L. and X.Y.; Software, L.C.; Supervision, C.L.; Validation, L.C. and H.H.; Visualization, L.C.; Writing—original draft, L.C. and X.Y.; Writing—review & editing, H.H. All authors have read and agreed to the published version of the manuscript.

**Funding:** The authors received General Projects of National Social Science Foundation (education) "internationalization development and overseas communication strategy of university brand image driven by education of foreign students in China" (BGA200057) for this research.

**Institutional Review Board Statement:** Not applicable.

**Informed Consent Statement:** Not applicable.

**Data Availability Statement:** This data can be provided by corresponding author on request.

**Acknowledgments:** The authors would like to extend our sincere thanks to anonymous reviewers for providing helpful comments and suggestions on earlier drafts of the manuscript.

**Conflicts of Interest:** The author declared no potential conflicts of interest with respect to the research, authorship, and/or publication of this article.

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
