# Peer review of "Research on Influencing Mechanism of Fashion Brand Image Value Creation Based on Consumer Value Co-Creation and Experiential Value Perception Theory"

_sustainability, doi:10.3390/su14137524_

Round 1

Reviewer 1 Report

The authors improve the article, following the reviewers remarks, with a new shape and new  arrangement.

Author Response

Comment 1: The authors improve the article, following the reviewers remarks, with a new shape and new arrangement.

Response to the reviewer: We thank the anonymous reviewer’s comment and confirmation! We appreciate the reviewer's suggestions during the previous revision round that resulted in such improvements. 

Reviewer 2 Report

The article is interesting, and the researched problem has scientific potential. However, some problems need to be solved:

 1. Introduction must contain a brief description of the sections of the paper.

 2. Data processing is performed using descriptive statistics. Therefore, the article will gain value if the author uses SEM to establish the relationships among variables. 

3. The discussion section should be built in the context of dialogue with researchers in the literature review. 

4. In my opinion, a section of conclusions that includes research limitations and future research directions would be helpful. 

The article presents the scientific value and can be published after carefully reviewing the reported issues.

Author Response

Comment 1: The article is interesting, and the researched problem has scientific potential.

Response: We thank the reviewer’s comment and confirmation!

Comment 2: Introduction must contain a brief description of the sections of the paper.

Response: We apologize for this insufficient description in the introduction. A brief description of the sections of the paper has been added to the revised manuscript.

Comment 3: Data processing is performed using descriptive statistics. Therefore, the article will gain value if the author uses SEM to establish the relationships among variables. 

Response: We thank the reviewer’s suggestions. The SEM model has been established in the revised manuscript.

Comment 4: The discussion section should be built in the context of dialogue with researchers in the literature review. 

Response: We agree with the reviewer, that it is good practice to bring literature to the discussion section or to build the discussion dialogue in the context of the literature review. However, section 2 was already devoted to the literature review and there was a chance of repletion of ideas thus we have avoided including it again.

Comment 5: In my opinion, a section of conclusions that includes research limitations and future research directions would be helpful

Response: We thank the reviewer’s suggestions. Research limitations and future research directions have been added to the revised manuscript.

Comment 6: The article presents the scientific value and can be published after carefully reviewing the reported issues.

Response: We thank the reviewer’s comment and support! The manuscript has been carefully revised according to the reported issues. We believe that the reviewer will consider it acceptable for publication.

Reviewer 3 Report

There are several limitations of the study a few of them are given below:

(1) There is a need to add a theoretical background that drives constructs used in the proposed model  

(2) Common method bias test needs to be reported

(3) Add mediation and moderation analysis with separate heading  

(4) There is a need to revise the discussion, and add the research consistent or inconsistent with the results.

(5) Sub-divided conclusion section into theoretical, practical, and limitation and future studies.

(6) Lastly, there is a need to add a pictorial form of moderation effect. 

Author Response

Comment 1: There are several limitations of the study a few of them are given below:

Response: We are thankful to the reviewer for devoting valuable time to giving valuable insights. We found the suggestions valuable and have revised the manuscript according to the comments.

Comment 2: There is a need to add a theoretical background that drives the constructs used in the proposed model.

Response: We thank the reviewer’s suggestions. The theoretical background that drives constructs used in the proposed model is consumer value co-creation theory and experiential value perception theory, which have been added in the revised manuscript.

Comment 3: Common method bias test needs to be reported

Response: We appreciate the suggestion of the reviewer, as it is understandable that all tests cannot be merged into one paper, thus we have avoided this test. We have rather choose the alternative tests and analyses that are closely related to the context of the research. We hope the reviewer would consider our apologies for not adding this test.

Comment 4: Add mediation and moderation analysis with separate heading  

Response: We thank the reviewer’s suggestions. The separate heading of mediation and moderation analysis has been added in the revised manuscript.

Comment 5: There is a need to revise the discussion, and add the research consistent or inconsistent with the results.

Response: We have revised the discussion accordingly.

Comment 6: Sub-divided conclusion section into theoretical, practical, and limitation and future studies.

Response: We thank the reviewer’s suggestions. Sub-divided conclusion section has been revised into theoretical-practical, limitation, and future studies.

Comment 7: Lastly, there is a need to add a pictorial form of moderation effect. 

Response: We thank the reviewer’s suggestions. The pictorial form of the moderation effect has been added in the revised manuscript.

Round 2

Reviewer 2 Report

The paper can be published in current form.

This manuscript is a resubmission of an earlier submission. The following is a list of the peer review reports and author responses from that submission.

Round 1

Reviewer 1 Report

Comment 1: The title should be changed according to the research results.

Comment 2: The authors need to argue the research gap in further detail in the introduction.

Comment 3: In prior studies, value creation is often built as the second-order construct; hence, the authors need to argue why this variable is selected as the first-order construct.

Comment 4: How to collect the data need to provide more details in terms of the collection period, data collection approach, and so on. Generally, the authors need to provide more evidence to convince the data collection was more reliable.

Comment 5: How did the authors solve the common method bias in the data collection. In other words, the authors need to test it.

Comment 6: The authors need to provide the discussion section. The authors need to compare the testing results with previous studies presented in the literature review.

Comment 7: The authors need to clarify the concepts of value creation and value co-creation.

Comment 8: The implications should withdraw from the findings. 

Author Response

We thank the reviewer for devoting valuable time to review our manuscript and for the helpful suggestions to make this manuscript better. We have addressed all the concerns expressed by the reviewers. Detailed responses to the comments by the reviewers are provided below.

Comment # 1

The title should be changed according to the research results.

Response to the reviewer

We agree with the reviewer’s comment. The title has been modified in the manuscript.

Comment # 2

The authors need to argue the research gap in further detail in the introduction.

Response to the reviewer

We apologize for this short description of the research gap. The research gap has been added to the revised manuscript.

Comment # 3

In prior studies, value creation is often built as the second-order construct; hence, the authors need to argue why this variable is selected as the first-order construct.

Response to the reviewer

We thank the reviewer’s question. The reason for value creation as a first-order construct is to explore the specific indicators of the value creation of fashion brand image from the micro-level based on case studies and literature studies. The research results will provide scientific evidence for fashion brands on creating value and from which aspects to develop brand image value.

Comment # 4

How to collect the data need to provide more details in terms of the collection period, data collection approach, and so on. Generally, the authors need to provide more evidence to convince the data collection was more reliable.

Response to the reviewer

We apologize for this insufficient explanation. We have added the statement accordingly.

Comment # 5

How did the authors solve the common method bias in the data collection. In other words, the authors need to test it.

Response to the reviewer

We thank the reviewer’s comments. We have added the statement accordingly. We believe the reviewer would find the revised manuscript considerable.

Comment # 6

The authors need to provide the discussion section. The authors need to compare the testing results with previous studies presented in the literature review.

Response to the reviewer

We agree with the reviewer’s comments. The discussion part has been built into the revised manuscript.

Comment # 7

The authors need to clarify the concepts of value creation and value co-creation.

Response to the reviewer

We apologize for the unclear descriptions. Brand value creation is a means for enterprises to establish a competitive advantage by relying on brand image and reputation with the power of the brand. Fashion brand value creation refers to a series of innovative creation activities produced by brands or enterprises to meet the higher target value pursuit and brand value maximization of customers. 

Value co-creation refers to consumer-led value creation activities, and the identity of customers is value co-creator. The leading value co-creation in the field of consumption is the consumer, and the value is expressed as the consumer’s experience value. In fact, consumer value co-creation can also be understood as customer participation in value co-creation.

Comment # 8

The implications should withdraw from the findings. 

Response to the reviewer

We agree with the reviewer’s comment. The implications have been modified in the revised manuscript.

Reviewer 2 Report

The paper is interesting and well done from statistical point of view.

Abstract: mention the target and where you apply the survey

In my opinion I suggest to make a separate  section which will include

  • the model Figure 1.
  • the 10 hypothesis  to be  more clearly for reader.
  • short presentation of survey .

Author Response

We thank the reviewers for devoting valuable time to review our manuscript and for the helpful suggestions to make this manuscript better. We have addressed all the concerns expressed by the reviewers. Detailed responses to the comments by the reviewers are provided below.

Comment # 1

The paper is interesting and well done from statistical point of view.

Response to the reviewer

We thank the reviewer’s comments and affirmation of the manuscript.

Comment # 2

Abstract: mention the target and where you apply the survey.

Response to the reviewer

We apologize for this mistake. The target and survey methods have been added to the revised manuscript.

Comment # 3

In my opinion I suggest to make a separate section which will include the model Figure 1. the 10 hypothesis to be more clearly for reader.

Response to the reviewer

We thank the reviewer’s suggestions. The separate section, including Figure 1, has been modified in the revised manuscript.

Comment #4

Short presentation of survey.

Response to the reviewer

We agree with the reviewer’s comment. The presentation of the survey has been added to the manuscript.

Reviewer 3 Report

Thanks to the authors for their contribution. But the manuscript has some problems to be addressed.

The Abstract should give expression to the necessity of the research.

The description of the research purpose in the Introduction is insufficient. The study of the relationship between brand image value and brand loyalty has been a relatively mature research field, but in this manuscript, I don't see the necessity of the research.

The number of the titles is chaotic so the table of contents needs to be reordered.

Although the manuscript indicates that the number of valid questionnaires was 567, it does not specify when and how the questionnaires were distributed and collected.

Does the study have a multicollinearity problem? Analysis data are necessary to figure it out.

A Discussion section is suggested to add, and the Conclusion should revolve around empirical analysis results. The Future Suggestion section is more like the authors’ subjective thoughts. For example, the authors believe that the primary way to increase the offline interactions between consumers and a brand is the celebrity effect, but this is not mentioned at all in the research design. Hence, where does this conclusion come from? In addition, if this study focuses on Chinese consumers, it had better be reflected in the title.

I would like to see the differences between this manuscript and existing ones as well as its innovations. Moreover, the Conclusion section needs to be reorganized.

Reviewer 4 Report

First of all, I would like to congratulate you on your paper. The article is interesting, and the researched problem has great scientific potential. However, some problems need to be solved:

  1. Please provide a graphical representation of the SEM model.
  1. In my opinion, a section of discussion should be built in the context of dialogue with researchers in the literature review.

The article presents the scientific value and can be published after carefully reviewing the reported issues.

Author Response

We thank the reviewer for devoting valuable time to review our manuscript and for the helpful suggestions to make this manuscript better. We have addressed all the concerns expressed by the reviewers. The detailed responses to the comments by the reviewers are provided below.

Comment # 1

First of all, I would like to congratulate you on your paper. The article is interesting, and the researched problem has great scientific potential. 

Response to the reviewer

We thank the reviewer’s comments and affirmation of the manuscript.

Comment # 2

Please provide a graphical representation of the SEM model.

Response to the reviewer

In order to answer the reviewer only and explain our manuscript more clearly to the reviewer, we have added a graphical representation of the SEM model into the attachment of these comments. However, we have not added it into the main text, considering the repetition of ideas, as all this has already been discussed in the manuscript, and adding this might confuse readers rather than clarify the main idea. We hope the worthy reviewer will agree with our point of view.

Comment # 3

In my opinion, a section of discussion should be built in the context of dialogue with researchers in the literature review.

Response to the reviewer

We agree with the reviewer’s comments. The discussion part has been built in the revised manuscript.

Comment # 4

The article presents the scientific value and can be published after carefully reviewing the reported issues.

Response to the reviewer

We thank the reviewer’s comments and affirmation of the manuscript. We have carefully modified the manuscript according to the reviewed reported issues.

Round 2

Reviewer 1 Report

Most of my previous comments have not been solved in the revision.

Reviewer 3 Report

Dear Authors, Thanks for revising the paper, I have read your paper several times just to be sure of my decision. But current manuscript still need to be improved, I cannot find the creation points distinguish from the previous studies.The paper must be significantly improved to meet any journal requirements. It is simply too early for your work to be published.